# Development and Validation of an Explainable Machine Learning-Based Prediction Model for Drug–Food Interactions from Chemical Structures

**DOI:** 10.3390/s23083962

**Published:** 2023-04-13

**Authors:** Quang-Hien Kha, Viet-Huan Le, Truong Nguyen Khanh Hung, Ngan Thi Kim Nguyen, Nguyen Quoc Khanh Le

**Affiliations:** 1International Ph.D. Program in Medicine, College of Medicine, Taipei Medical University, Taipei 110, Taiwan; 2AIBioMed Research Group, Taipei Medical University, Taipei 110, Taiwan; 3Department of Thoracic Surgery, Khanh Hoa General Hospital, Nha Trang City 65000, Vietnam; 4Department of Orthopedic and Trauma, Cho Ray Hospital, Ho Chi Minh City 70000, Vietnam; 5Undergraduate Program of Nutrition Science, National Taiwan Normal University, Taipei 106, Taiwan; 6Professional Master Program in Artificial Intelligence in Medicine, College of Medicine, Taipei Medical University, Taipei 110, Taiwan; 7Research Center for Artificial Intelligence in Medicine, Taipei Medical University, Taipei 110, Taiwan; 8Translational Imaging Research Center, Taipei Medical University Hospital, Taipei 110, Taiwan

**Keywords:** adverse food reaction, chemical informatics, drug–food interactions, drug–nutrient interactions, DrugBank, explainable artificial intelligence, FooDB, machine learning, precision medicine, simplified molecular-input line-entry system

## Abstract

Possible drug–food constituent interactions (DFIs) could change the intended efficiency of particular therapeutics in medical practice. The increasing number of multiple-drug prescriptions leads to the rise of drug–drug interactions (DDIs) and DFIs. These adverse interactions lead to other implications, e.g., the decline in medicament’s effect, the withdrawals of various medications, and harmful impacts on the patients’ health. However, the importance of DFIs remains underestimated, as the number of studies on these topics is constrained. Recently, scientists have applied artificial intelligence-based models to study DFIs. However, there were still some limitations in data mining, input, and detailed annotations. This study proposed a novel prediction model to address the limitations of previous studies. In detail, we extracted 70,477 food compounds from the FooDB database and 13,580 drugs from the DrugBank database. We extracted 3780 features from each drug–food compound pair. The optimal model was eXtreme Gradient Boosting (XGBoost). We also validated the performance of our model on one external test set from a previous study which contained 1922 DFIs. Finally, we applied our model to recommend whether a drug should or should not be taken with some food compounds based on their interactions. The model can provide highly accurate and clinically relevant recommendations, especially for DFIs that may cause severe adverse events and even death. Our proposed model can contribute to developing more robust predictive models to help patients, under the supervision and consultants of physicians, avoid DFI adverse effects in combining drugs and foods for therapy.

## 1. Introduction

The intended efficiency of medications or therapies can be altered due to drug–food constituent interactions (DFIs) [1]. While most DFIs cause no harm or mild effects, some may lead to adverse drug events (ADEs) [2] or severe consequences. Therefore, the awareness of DFIs is pivotal for the safe usage of oral drugs, food types, and food-originated supplements to ensure the efficacy of therapies. As the number of prescriptions containing drug–drug combinations rockets [3,4], the adverse DFIs experience a corresponding rise. The bioavailability and other parameters of medications could be reduced or increased due to the interactions with different food types or even with the physiological secretion of the body itself, e.g., the secretion of gastric acid, pancreatic digestive chemicals, and more [1]. More than 30 percent of the reported adverse drug events are related to DFIs [5]. Moreover, one of the most common causes of drug or medicament withdrawals from the market is owing to DFIs [6,7,8]. In contrast, the figure for the usage of any prescriptions rose by nearly five percent to reach 85% in 2012 [3]. This can exacerbate the probability of DFI since clinicians or patients can ensure no interactions between oral drugs and food intake.

Verapamil, terfenadine, nifedipine, cyclosporine, and more drugs were also reported with adverse events when taken with grapefruit juice [9]. It is due to the inhibiting effects of grapefruit juice on intestinal *CYP3A4*, a crucial isoenzyme of cytochrome *P450* helping eliminate drugs [10]. Warfarin was reported to interact with various food types, in which altering the anti-coagulant effect is the most critical consideration [11,12,13]. Alcohol could cause severe adverse events, and even death, when combined with diazepam, lorazepam, acetaminophen, or methotrexate [14,15,16,17,18,19]. Foods rich in [K+], such as oranges and bananas, may cause hyperkalemia, resulting in cardiac arrest and death due to myocardial arrhythmia. Therefore, they must not be taken with renin-angiotensin system inhibitors since they elevate the blood potassium concentration by reducing aldosterone activity. In addition, tyramine-rich foods (e.g., wine, cheese, etc.) must not be combined with monoamine oxidase inhibitors (MAOIs), which are used for patients with depression. The consequence is the onset of a hypertensive crisis resulting from the growth of catecholamine biosynthesis caused by the breakdown reduction of tyramine [20].

There is a dire need to stratify which drug–food compound pairs can be taken together for the safety of patients during therapy [21]. A study in 2004 indicated that DFIs, not only among foods but also in dietary supplements, were responsible for mitigating medications’ efficacy or lasting pharmacological effects, negatively affecting patients’ health and drug labeling [6]. Four hundred sixty-two medicinal products were withdrawn from the market due to adverse drug reactions in DFIs and drug-drug interactions (DDIs) [7]. It is also evident that the elderly suffered the most from harmful DFIs [22]. Aging patients, associated with chronic comorbidities, hygiene, and changes in physiological processes, require multiple-drug therapy. One study on the senior Brazilian population [23] has depicted the correlation between disease conditions, metabolism status among aged people, and risks of adverse events resulting from DFIs. Furthermore, Mason [24] revealed that the risks of DFIs increased among infants, children, and patients who have undergone long-term therapy or those in need of particular nutritional demand (cancer, burns, diarrhea, etc.).

Various strategies have been proposed to study the adverse interactions between drugs and food constituents. Mallet et al. [25], and Chan [26] provided some measures to treat unexpected DFIs. However, the first-step clinical approach to avoiding DFIs is carefully investigating which ingesting food can cause interactions with administering oral medications and planning an appropriate combination of drugs and dietary supplements or food intake. Jensen et al. [27] proposed a MEDLINE database to explore DFIs to improve therapeutic intervention strategies. Their findings concentrated on the interactions between plant-based food constituents and drug targets. Another approach introduced by Riedmaier et al. [28] integrated with in vitro data with the knowledge of human physiology to predict food effects on drug compounds. This framework has curbed the drawbacks of other previous methods using only in vitro samples [29,30,31], or animal experiments [31,32,33]. Qin et al. deployed spectroscopy and molecular interaction to discover the inhibitory effects of apigenin and luteolin on advanced glycation end products in bovine serum albumin (BSA)-glucose and BSA-fructose models [34], in which Auto dock software (version 4.0, Scripps Institute, USA) was used to compute the minimum binding energy of the investigated proteins. However, the methods above required much information on drug targets, interacting foods, and particular effects, which usually miss input data to classify the DFIs.

The recent development of machine learning (ML) in drug discovery [35] has facilitated DDI classification tasks [5,22,36,37]. No DFI study applied advanced ML techniques to differentiate between the interaction effects of drug and food compounds, except Ryu et al. [5] and Rahman et al. [38]. In Ryu et al. [5], albeit indirectly, they employed their deep learning network trained on DDI data to discriminate the DFIs. With the presence of simplified molecular-input line-entry system (SMILES) formulas represented for constituents and public databases (e.g., DrugBank [39] and FooDB (https://foodb.ca/, accessed on 29 August 2022), Ryu et al. showed the potential of ML implementation in identifying DFIs. However, there was a need to assemble a model that could detect various DFIs straightforwardly without relying on this. Rahman et al. [38] applied a graph mining approach to forecasting DFIs. Despite the novelty, the experiment was complex to reproduce under many conditions, and the final results were unsatisfactory. Moreover, the method did not cover all drug–food constituents.

Herein, we introduced a predictive model, which utilized simple ML algorithms to classify the DFIs more conveniently, accurately, and efficiently based on their SMILES. Input data are the SMILES of drug and food compounds, which express their codified chemical structures [40]. Our model was built on an eXtreme Gradient Boosting (XGBoost) classification algorithm, with the eighteen most essential features refined through a tight, four-step feature selection method. We evaluated the robustness of our model’s prediction on one external test set. Furthermore, we used our predictor to recommend the combination of some drugs and foods in clinical practice. Clinical physicians could use the results interpreted from the model since the interactions written in readable sentences could contribute to drug prescriptions and dietary decisions.

## 2. Materials and Methods

The workflow of our study, which is exhibited in Figure 1, comprised three key steps: (i) Data gathering and pre-processing, (ii) Model building and optimization, and (iii) Validations and recommendations. In the first step, we extracted the data from DrugBank [39] (version 5.1.7) and FooDB (https://foodb.ca/, accessed on 29 August 2022) (version 1.0) databases. During this step, we processed the data to reduce the number of drug and food constituents examined to 1133 and 4341, respectively. In the second step, we applied four feature selection methods to pick up eighteen optimal features among 3780. We applied five algorithms to the training set with eighteen features and then fine-tuned the parameters to find the best model. The last step evaluated the performance of our model on one external test set from the previous study [5]. Finally, we extended our study to predict some adverse DFIs of various drugs encountered in clinical practice.

### 2.1. Data Gathering and Pre-Processing

#### 2.1.1. Data Gathering

We primarily collected data from the DrugBank (version 5.1.7, 2020) [39] and FooDB (version 1.0, 2020) databases, which contained 13,580 drugs and 70,477 food constituents, respectively. We focused on DFI annotations in DrugBank and consistently selected only canonical SMILES for food and drug compounds [40].

#### 2.1.2. Data Pre-Processing

In the FooDB database, we eliminated food compounds without SMILES (N = 70) or duplicates (N = 809). We then excluded similar foods using Tanimoto’s coefficient, which calculates the structural similarity between two constituents [5,41]. First, we used one compound as a “target” and then matched that compound to the other “query” molecules in the dataset to calculate the coefficients. This step was repeated until the last pair of compounds in the dataset. Second, we rejected which food compounds were similar in structure, meaning they had a structural similarity coefficient greater or equal to 0.75 [5,41,42]. This process resulted in the removal of 65,257 food compounds. Finally, the number of food compounds for further analysis was 4341.

For the drug database (N = 13,580), we removed compounds that do not have information on the SMILES (N = 2889) and/or do not have annotations on interactions for food compounds (N = 12,395). The final drug data set contained 1133 drug compounds with SMILES formulas and annotations of DFIs.

#### 2.1.3. Labeling of DFIs Ground Truth

Based on annotations from DrugBank, we categorized the DFIs into three groups:Positive interactions (Class 1): (1) if two of the following conditions are met simultaneously. (a) If a drug compound in combination with a food compound, and the food compound increases the extent of absorption, increases bioavailability, increases peak concentrations, and decreases time to peak concentrations of the drug; (b) no adverse effect or toxicity for human health has been observed from DrugBank annotations. (2) If the DrugBank annotations indicate that when the drug compound is taken with the food compound, the food compound will benefit the patient (e.g., food reduces irritation, take with food to reduce nausea, food decreases the risk of gastrointestinal side effects, etc.) despite not specifying the interaction information in terms of pharmacokinetics, the pharmacodynamics of the drug. (3) If DrugBank annotations indicate “take after meals”, “take after a meal”, or “take with food” although they do not specify the information on pharmacokinetics, pharmacodynamics, or patient benefits when taking that drug with food compounds.Negative interactions (Class 0): (1) if two of the following conditions are met simultaneously. (a) If the drug is taken with food, but food reduces the extent of absorption, reduces bioavailability, decreases peak concentrations, and increases time to peak concentrations of the drug; (b) at least one adverse effect or toxicity for human health has been described from DrugBank annotations. (2) If the DrugBank annotations indicate that when the drug compound is taken with the food compound, the food compound will cause harm to the patient regardless of the interaction information in terms of pharmacokinetics and the pharmacodynamics of the drug. If DrugBank annotations contain the words “avoid”, “Take separately from meals”, “take on an empty stomach” or “take before a meal” regardless of the information on pharmacokinetics, pharmacodynamics, or benefits when taking that drug with food compounds.Non-significant interactions (Class 2): (1) if DrugBank annotations do not fall into the above two categories. (2) If DrugBank annotations state “take with or without food”, “take consistently with regard to food” regardless of pharmacokinetic or pharmacodynamic interaction information.

Drug compounds with the annotation “take at the same time every day” were not considered; however, if there were other annotations for the same drug that satisfied the conditions of the three groups mentioned above, the drug–food interactions were classified into the corresponding category. In detail, from 4341 food and 1133 drug compounds, there were 2,382,871 DFIs, including 476,642 negative DFIs, 776,146 positive DFIs, and 1,130,083 non-significant DFIs.

### 2.2. Model Building and Optimization

#### 2.2.1. Feature Extraction

We ran our experiments on Windows 10 (version 20H2) (4.60 GHz Intel i7-11800H CPU and 64 GB RAM). We used PyBioMed package [43] (PyInteraction module) and RDKit (version 1.0.3) [44] for input representation preparation of the chemical compounds. All calculations were implemented in Python 3.9.12. We extracted all drug–food pairs in SMILES to 2,382,871 number arrays with 3780 molecular operating environment (MOE) descriptors. MOE descriptors are features derived from each pair of DFIs, calculated from the low energy conformations of the composite.

*slogP* is the logarithm of octanol:water regional coefficient; this was calculated by Wildman and Crippen in 1999. *MTPSA* or “Molecule polar surface area” was calculated by estimating the polar surface area from the connection plane’s information, which was first introduced by Ertl et al. [45]. *VSA* was defined as the van der Waals surface area, which the polyhedral depiction of individual atoms could quantify. *PEOE* is the abbreviation of Partial Equalization of Orbital Electronegativities, which was introduced by Gasteiger et al. [46], and is a method used to calculate the atomic partial charges [46]. The Lorentz–Lorentz equation defines the molecular refractivity (*MR*) parameter or molecular refractivity:(1)$$MR=\left(\frac{n^{2}-1}{n^{2}+2}\right)\left(\frac{MW}{d}\right)$$
where *n* is the index of refraction, MW is the molecular weight, and *d* is the density. MR is strongly correlated with the polarization of the respective molecule. MR can also be estimated from group-relative constants when exploratory values are absent. In total, there were seven families of descriptors.

#### 2.2.2. Feature Selection

We performed feature selection using variance threshold (VT), which removes the features with variance lower than a threshold value of 0.8. The remained features would next undergo Pearson’s correlation coefficient feature selection. If two features correlate more than 75%, we will reject one that contributes less to the outcome. All features obtained from the previous step would then be passed to the least absolute shrinkage and selection operator (LASSO) [47], with alpha set to 0.001. Afterward, we performed the RidgeCV-based recursive feature elimination with the cross-validation (RFECV) technique (five iterations of cross-validation) to filter out the best features. There were 18 optimal features; the detailed names of all features can be found in the Section 4.

### 2.3. Model Training

Random forest (RF), extreme gradient boosting (XGBoost), extra trees (ET), light gradient boosting machine (LGBM), and one neural-network-based classifier multilayer perceptron (MLP) were five-time cross-validated on the training set to measure the baseline performance [48]. Next, all baseline models were fine-tuned using Randomized Search Cross-Validation (CV) (five iterations) on the validation set. The best classification model was chosen based on the performance of all models on the testing set. The CV and fine-tuning results can be found in the Section 4 (Table 1, Table 2 and Table 3).

### 2.4. Validation and Recommendations

For the final stage, we aimed to create a model capable of yielding recommendations or warnings to assist patients and physicians in using drugs in combination with food to avoid adverse effects. The user(s) input the names of the drug and food compound into the model. Then, the model will predict whether they can be used together or not. Based on that output, the user(s) will consult a physician for the best advice. If the DFI were positive, the predicted result would be “A could be taken with food containing B”. If the DFI were non-significant, the output would be “A may be taken with food containing B”. Otherwise, the recommendation would be “A should not be taken with food containing B”.

### 2.5. Evaluation Metrics

Four measurement metrics [49] were taken into account to assess the performance of each algorithm in classifying the interactions:(2)$$Accuracy=\frac{TP+TN}{TP+TN+FP+FN}$$
(3)$$Recall=\frac{TP}{TP+FN}$$
(4)$$Precision=\frac{TP}{TP+FP}$$
(5)$$F1-score=\frac{2\times Precision\times Recall}{Precision+Recall}$$

TP, TN, FP, and FN represent True positive, True negative, False positive, and False negative, respectively. For each class in the multi-classification task, the Accuracy of each class is the ratio of correctly predicted instances of that class and the total number of instances. Recall is the fraction between the number of accurately predicted DFIs and all positive DFIs. Precision is calculated by the division of accurately classified DFIs and total DFIs. Recall and Precision are related; increasing the former may decline the latter, and vice versa. Therefore, the F1-score, the geometric mean of Recall and Precision, makes the result interpretations more plausible.

## 3. Results

### 3.1. 4341 Food Compounds and 1133 Drug Compounds from DrugBank and FooDB

Of the 4341 food compounds, 20 were inorganic, 1095 were organic, and the remaining 3226 were missing information. Among 1133 drug compounds, 22 were inorganic, 1063 were organic, and 48 were unknown. We excluded the drug–food pairs having the same mean value of 3780 features. This helps us to avoid training the models repetitively with similar interacting pairs and to increase the generalizability. This process reduced the number of samples from 2,382,903 in the benchmark dataset to 2,263,474. We then randomly split 1,133,652 (50.08%) for training, 848,803 (37.50%) for hyperparameter tuning (the validation set) and 281,019 (12.42%) for testing (the internal test set).

For the external test set, we extracted DFIs from a previous study by Ryu et al. [5]. They applied a predictive model to a validation set of 4567 DFIs. We removed 2319 duplicates of drug–food pairs, 66 drugs and 260 foods with no SMILES data in DrugBank and FooDB, respectively. Finally, the external test set consisted of 1922 instances, with 751, 378, and 793 pairs of negative, positive, and non-significant DFIs, respectively. More importantly, it contained 232 drugs not included in the training data.

### 3.2. 18 Selected Features Can Improve the Prediction

With the threshold set to 0.8, VT eliminated 935 features from 3780 original features. For the remaining 2845 features, we normalized all training data using the standardization method from *scikit-learn* and performed Pearson’s correlation coefficient feature selection process. After the experiment, we picked up 1568 features correlating more to the classification outcome. Next, we fed 1568 features to LASSO with an alpha of 0.01 and filtered out 52. Finally, we performed the RidgeClassifierCV-based RFECV (cross-validation set to 5, performed on five iterations) to obtain 18 features, including *MTPSA+MTPSA, MRVSA9, MRVSA8, MRVSA0, MRVSA2, VSAEstate10+VSAEstate10, EstateVSA0*LabuteASA, PEOEVSA12, PEOEVSA10, PEOEVSA5, PEOEVSA9, slogPVSA2, slogPVSA0, slogPVSA9, VSAEstate7+VSAEstate7, EstateVSA7, EstateVSA2, EstateVSA1*VSAEstate8*.

We applied four ML algorithms and one neural network architecture to our training set: RF, XGBoost, ET, LGBM, and MLP. For each classifier, we implemented a five-fold CV to evaluate their performances. In Table 1, XGBoost reported the highest performance, achieving a mean accuracy of 96.75% (±0.05%). The lowest accuracy was seen on MLP (95.89% ± 0.02%), whereas LGBM, RF, and ET gave a score ranging from 95.89% to 96.71%.

### 3.3. Performance Improvement Via Hyper-Parameter Tuning

We applied randomized search cross-validation to tune the optimal parameters for all five algorithms. Six parameters were adjusted (Table 2). Hyper-tuned parameters improved the performance of the baseline XGBoost model by 0.0004 (Table 3), gaining the highest accuracy of 96.77%. The tuned XGBoost model was afterward used to predict DFIs in the external test set (Table 4 and Figure 2).

### 3.4. Evaluating the Performance Results of the Final Models on External Test Set

We evaluated the model’s classification performance on the internal validation set and the external test set (which was based on drugs and foods used in the study of Ryu et al. [5]). The external test set consisted of 377 drugs and 59 food compounds that comprised 1922 drug–food pairs, with 751, 378, and 793 pairs of negative, positive, and non-significant DFIs, respectively. The external test set contained 232 drugs without the annotations of interactions from our database. Our model correctly predicted 97.56% of drug–food pairs (Table 4 and Figure 2), showing the model’s predictive capacity on unseen data.

### 3.5. Interpretation of Eighteen Optimal Features

*MRVSA0* is calculated using *MR* and surface area contributions. *MRVSA0* ranked first in the feature importance list. *PEOEVSA5* and *PEOEVSA9* compute the sum of one atom’s van der Waals surface areas where its partial charge is between 0.25 and 0.30. *EstateVSA7* and *EstateVSA2* [43] are computed using Estate indices and surface area contributions. *slogPVSA2* denotes the relative approachable van der Waal’s surface area, computed for each atom with a contribution to the partition coefficient log (octanol/water) in the range of (−0.2, 0) [50]. Its negative coefficient value indicated that high hydrophobicity decreased the agonist activity.

According to the SHAP (SHapley Additive exPlanations) [51] analysis in Figure 3, the x-axis portrays the SHAP value calculated for each feature based on their contribution to the final output, while the y-axis depicts the feature names. In this study, SHAP values greater than zero (i.e., the right side of the x-axis) indicate that the model predicts non-significant DFIs, while SHAP values less than zero (i.e., the left side of the x-axis) indicate negative DFI predictions. SHAP values close to zero indicate positive DFI outputs. We can observe that the over-zero side of the x-axis is longer than the opposite one because of the larger number of non-significant DFIs compared to other groups. Each point on the chart represents one SHAP value for a prediction and feature. The red color indicates a higher value of a feature, while blue indicates a lower value of a feature.

Accordingly, the red dots of *MRVSA0, EstateVSA2, MRVSA9, MRVSA8* and blue dots of *PEOEVSA5, MTPSA+MTPSA, VSAEstate10+VSAEstate10* gather on the right side of the x-axis, indicating that the high values and low values of these features, respectively, direct the model in recognizing the non-significant DFIs. High *PEOEVSA5, EstateVSA7, slogPVSA9, MTPSA+MTPSA*, and low values of *MRVSA0, MRVSA9* help detect the negative DFIs. The positive DFIs are identified by the increasing values of *PEOEVSA5, EstateVSA0*LabuteASA, EstateVSA1*VSAEstate8* and the decline of *PEOEVSA9, EstateVSA7, EstateVSA2, slogPVSA9, MRVSA2, VSAEstate7+VSAEstate7, slogPVSA0, PEOEVSA12*.

The polarization and van der Waals surface area of the interacting molecules seem to have a critical role in determining whether a drug–food constituent combination is negative or non-significant. At the same time, they cannot assist in distinguishing a positive one. This can be observed in the rise or fall of *MRVSA0* and *MRVSA9*. Furthermore, interacting atoms with higher “octanol:water” regional coefficient, polar surface area, and van der Waals forces tend to generate negative drug–food constituent pairs. On the other hand, molecules with high electrotopological state indices combined with increasing van der Waals surface area and orbital electronegativities may indicate a positive interaction between respective drugs and foods. These observations are drawn from the model output and, thus, are needed to be confirmed by biological experiments.

### 3.6. The Interpretation of our Model to Clinical Physicians, Pharmacists, and Patients

We input nine commonly used drugs and seven food compounds known to cause adverse reactions to test the model’s predictive power in clinical practice. We focus on negative interactions because we believe they are more clinically significant than positive and non-significant interactions. Additionally, we have attached scientific evidence about adverse interactions explaining why the listed pairs should not be used together for reference (see Table 5).

## 4. Discussions

The acknowledgment of DFIs is currently underestimated compared to the concerns for DDIs. The lack of methods to classify DFIs, drug–nutrient interactions (DNIs), and further predict novel interactions contributes to the massive manufacturing of drugs regardless of underlying harmful inter-activities towards food and dietary plans. Thus, we propose an ML model capable of predicting the directions of compound-level interactions using SMILES structures.

After applying Tanimoto’s coefficient, the number of food compounds could raise concerns about insufficient food compounds, potentially reducing the model’s predictive power for unseen patterns. However, we believe that applying the Tanimoto coefficient helps to increase generalizability, meaning that 4,341 food constituents can represent others. To test our model’s ability to predict negative DFIs, we looked at Geranyl rhamnosyl-glucoside (not included in the final food dataset) to assess whether our model could predict a negative interaction (“Avoid alcohol”) with Nitroglycerin. The model predicted that “Nitroglycerin should not be taken with Geranyl rhamnosyl-glucoside”, which was true. In the second example, the model correctly predicted that Warfarin and Dicoumarol should not be taken together, even though Dicoumarol was not included in the 4341 foods.

In this study, we used DrugBank annotations to label drug–food interactions as “positive”, “negative”, and “non-significant”. In the literature, most studies classify drug–food interactions based on pharmaceuticals–pharmacokinetics–pharmacodynamics [1,23,65]. However, no study has provided a straightforward and convenient way to apply ML. Therefore, our classification model can pave the way for further studies on the same topic.

Our proposed framework, with four stages of refinement, removes similar features and reduces collinearity to improve the model’s performance. From 3780 features, we used various feature selection methods to sort out the most significant ones for the outcome. Eventually, we selected only the five most important features, each representing different aspects of drug or food constituents (see Figure 3). XGBoost has long been used as an efficient algorithm for classification problems. Its simplicity, high stability, scalability, and ability to prevent overfitting [66] make XGBoost a robust classifier, particularly in high-dimensional datasets. Hypertuning further boosted the performance of XGBoost, demonstrating the potential of this classifier algorithm to correctly detect different types of DFIs. The experimental results also revealed that XGBoost outperformed MLP, a neural network-based model, showing that XGBoost can perform better on this type of data.

As previously mentioned, we computed interactions between drugs and food constituents from SMILES. This is similar to previous research conducted by Ryu et al. [5] and Rahman et al. [38]. However, our study differed from Ryu et al. [5] in terms of labeling the data and using a predictive model. In our study, we directly predicted drug–food interactions from the SMILES formula of drug–food constituents. In contrast, Ryu and colleagues predicted drug–food interactions based on similarity with drug–drug interactions using a Tanimoto coefficient of more than 0.75 (Table 6). However, this approach may not cover all food molecules, as not all foods are structurally similar to available drugs. Additionally, their performance on drug-food data was not mentioned, making a complete comparison impossible.

Similar to our research, Rahman et al. [38] used the SMILES of food directly as input data and based their study on the same databases. However, their training set contained only 788 drugs for analysis. Although they gathered more food constituents (16,230 foods), their selected foods did not represent the whole FooDB database compared to our work, since we used Tanimoto’s coefficient for structural similarity filtration. Moreover, our model can detect drug–food interactions better than Rahman’s approach, with precision ranging from 0.9265 to 1.0 on the external test set compared to their model’s highest precision of 0.84 (Table 6). Our model’s simplicity is also superior to their proposed method regarding reproducibility.

Knowing which drugs to combine with which food compounds is essential for managing dietary intake plans and improving treatment outcomes. Based on the model results, we aim to extend this study to provide drug–food combination recommendations. To test this, we examined seven commonly used clinical drugs and eight foods that should not be combined according to DrugBank annotations. For example, consuming alcohol during treatment with Methotrexate may lead to hepatotoxicity [16,54,55], while the use of grapefruit juice during treatment with nisoldipine may cause toxicity to patients [59]. Our model successfully recommended that Methotrexate and nifedipine should not be taken with these food compounds. However, it may take some time to improve this extension, and we hope that it will become more helpful in the future.

Despite the promising results of our model, various limitations remained and needed to be addressed. The first drawback to be considered is the shortage of databases due to invalid information, such as origins, class, and interactions towards broader classes of different compounds, of food, nutrients, and drug constituents. In the future, with broader contributions to the public food database, we believe that ML models’ ability to predict DFIs will be even more impressive. Second, since there was no standard to determine in which situation a DFI would be positive or negative, the way we labeled data in this study, based on DrugBank annotations, may be inappropriate. Third, the outputs of our model were imperfect, as there were still many incorrect recommendations, which can negatively affect clinical practice. Furthermore, we can only distinguish between positive, negative, or non-significant results, but we cannot explain how negatively the DFIs affect our health. Many aspects need to be improved before our model can become a robust assistant for medical staff. Therefore, we emphasize that the recommendations of physicians or pharmacists are of the highest importance, and the outcomes from our model are only for further consideration. Fourth, although we proved that XGBoost was superior to a neural network algorithm, we still need to consider other deep methods. In 2018, Ryu et al. successfully proposed a cutting-edge deep neural network-based model to predict DDIs [5]. Recently, Lin et al. [67,68] proposed two state-of-the-art deep neural networks to discriminate DDIs. Motivated by this work, our future focus will be on recruiting deep learning techniques [69] to classify DFIs more accurately. Finally, the efficacy and stability of our model require further assessment of different databases, institutions, and contributions from worldwide scientists to complete the prediction model in the future.

## 5. Conclusions

To reduce the number of ADEs due to DFIs and DNIs, we propose a new classification model based on the XGBoost classifier and eighteen optimal features. The ability of our model to predict adverse interactions between drug-food compounds can contribute to drug discoveries and the conformity between prescribed drugs and dietary plans in clinical medicine. From a large number of drug and food constituents (13,580 and 70,477, respectively), we reduced them to only 1133 drugs and 4341 food compounds. We also identified the eighteen most important features that yielded the highest predictive performance on DFIs. Each feature represents a chemical reaction between atoms. Thus, our findings could contribute to understanding DFIs at the atomic level along with biological experiments. We believe these findings will benefit the scientific community and patients by decreasing the number of ADEs caused by DFIs and setting a new standard for DFI prediction models.

## Figures and Tables

**Figure 1 sensors-23-03962-f001:**
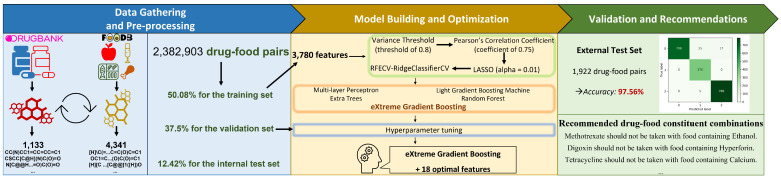
The workflow of our study. First, we obtained the SMILES notations of drug and food constituents from DrugBank and FooDB databases. After pre-processing, we filtered out 1133 drugs and 4341 food compounds, making 2,382,903 drug–food pairs in the benchmark dataset. We subsequently used *PyBioMed* and *RDKit* packages in Python to extract 3780 features of each interacting drug–food pair. We applied a four-step feature selection process to the training set to find the 18 most important features. Five classification algorithms were applied to the training data via five-fold cross-validation. As XGBoost gave the best prediction outcome, we fine-tuned it using the validation set. Finally, we tested our optimum XGBoost model on the internal test set and one external test set containing 1922 drug-food pairs. Finally, we used the model to recommend some common drug–food compound combinations.

**Figure 2 sensors-23-03962-f002:**
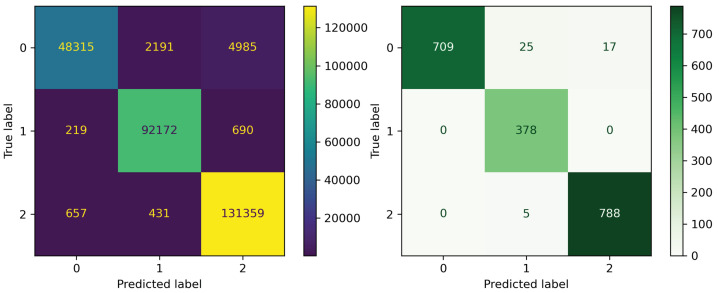
Confusion matrix of our optimal XGBoost model on the testing and the external test sets. On the testing set (**left** plot): The model most accurately detected positive and non-significant DFIs (recall 0.99 in both classes) while only recognizing 87% of negative DFIs. Likewise, on the external test set (**right** plot), the model recognized all positive DFIs and 99% of non-significant DFIs. Negative DFIs were recognized as acceptable, with 94% of those discriminated against.

**Figure 3 sensors-23-03962-f003:**
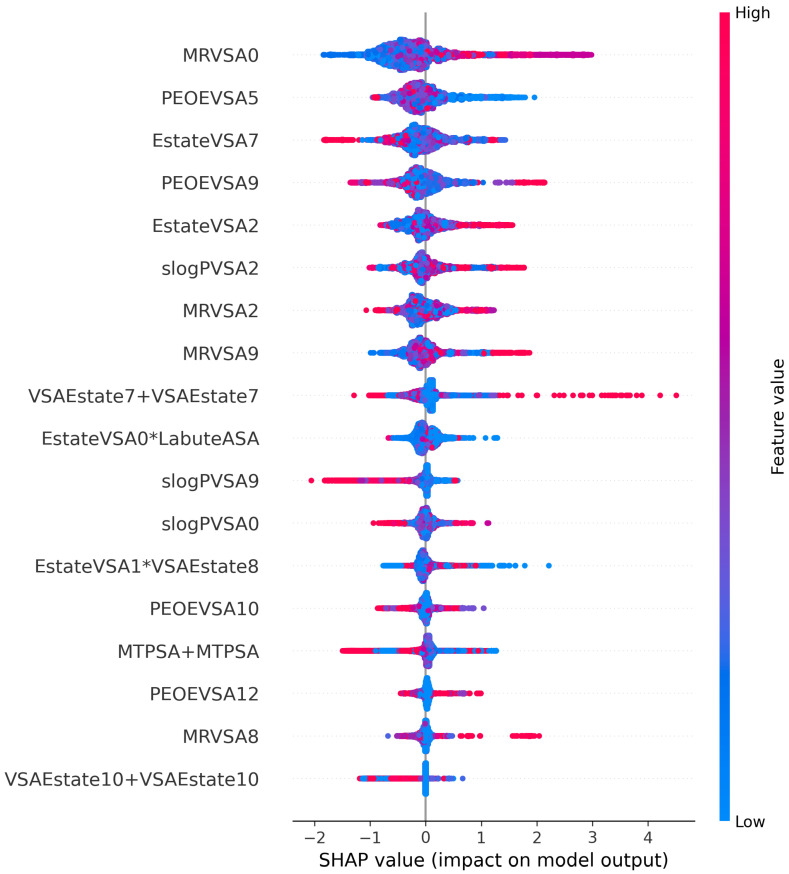
The SHAP (SHapley Additive exPlanations) plot of eighteen optimal features. The red dots of *MRVSA0, EstateVSA2, MRVSA9, MRVSA8* and blue dots of *PEOEVSA5, MTPSA+MTPSA, VSAEstate10+VSAEstate10* gather on the right side of the x-axis, indicating that the high values and low values of these features, respectively, direct the model in recognizing the non-significant DFIs. High *PEOEVSA5, EstateVSA7, slogPVSA9, MTPSA+MTPSA*, and low values of *MRVSA0, MRVSA9* help detect the negative DFIs. The positive DFIs are identified by the increasing values of *PEOEVSA5, EstateVSA0*LabuteASA, EstateVSA1*VSAEstate8* and the decline of *PEOEVSA9, EstateVSA7, EstateVSA2, slogPVSA9, MRVSA2, VSAEstate7+VSAEstate7, slogPVSA0, PEOEVSA12*.

**Table 1 sensors-23-03962-t001:** Performance results among different algorithms on training data (Five-fold cross-validation).

Classifier	Accuracy	STD
XGBoost	0.9675	0.0005
LGBM	0.9671	0.0004
RF	0.9651	0.0002
ET	0.9589	0.0004
MLP	0.9561	0.0050

STD: standard deviation; XGBoost: eXtreme Gradient Boosting; LGBM: Light Gradient Boosting Machine; ET: Extra Trees; MLP: Multi-Layer Perceptron.

**Table 2 sensors-23-03962-t002:** Default and tuned values of parameters used in our study.

Parameter	Default Setting	Hyper-Tuned Values	Optimal Value
n_estimators	100	[50, 100, 150, 200]	150
max_depth	6	[3, 4, 5, 6, 8, 10, 12, 15]	6
gamma	0	[0.0, 0.1, 0.2, 0.3, 0.4]	0.4
colsample_bytree	1	[0.3, 0.4, 0.5, 0.7]	0.3
min_child_weight	1	[1, 3, 5, 7]	5
learning_rate	0.1	[0.05, 0.1, 0.15, 0.2, 0.25, 0.3]	0.2

**Table 3 sensors-23-03962-t003:** Performance of classification algorithms before and after hyper-parameters tuning.

Algorithms	Before Tuning	After Tuning
XGBoost	0.9673	0.9677
MLP	0.9589	0.9623
LGBM	0.9671	0.9673
ET	0.9586	0.9611
RF	0.9651	0.9662

**Table 4 sensors-23-03962-t004:** The performance (recall, precision, and F1-score) of our optimal XGBoost model on the internal and external test sets.

Types of DFIs	Internal Test				External Test			
	Accuracy	Recall	Precision	F1-Score	Accuracy	Recall	Precision	F1-Score
Negative DFIs	0.9714	0.8707	0.9822	0.9231	0.9781	0.9441	1.0	0.9712
Positive DFIs	0.9874	0.9902	0.9723	0.9811	0.9844	1.0	0.9265	0.9618
Non-significant DFIs	0.9759	0.9918	0.9586	0.9749	0.9886	0.9937	0.9789	0.9862

**Table 5 sensors-23-03962-t005:** Interpretation of our XGBoost-based model to clinical practice.

Drug–Food Compound	Adverse Effect(s)	Model’s Recommendation	References
Tetracycline + Calcium	Calcium reduces the absorption rate of Tetracycline.	Tetracycline should not be taken with food containing Calcium.	Neuvonen et al. [52], Kuang et al. [53]
Methotrexate + Ethanol	Ethanol increases the risk for liver damage while taking Methotrexate.	Methotrexate should not be taken with food containing Ethanol.	Whiting-O’Keefe et al. [54], Price et al. [16], Malatjalian et al. [55], Humphreys et al. [14]
Diazepam + Ethanol	Ethanol may increase the central nervous depressant effect of diazepam, leading to dizziness, nausea, lost of consciousness, even coma, or death.	Diazepam should not be taken with food containing Ethanol.	Koski et al. [17,18]
Nitroglycerin + Ethanol	Drinking alcohol while taking this medication increases the risk for dangerously low blood pressure and Disulfiram-Like Reactions.	Nitroglycerin should not be taken with food containing Ethanol.	Weathermon et al. [56]
Digoxin + Hyperforin	St. John’s wort may decrease levels of the medication and reduce its effectiveness. Hyperforin is a natural compound extracted from the St. John’s wort (Hypericum perforatum) plant.	Digoxin should not be taken with food containing Hyperforin.	Johne et al. [57]
Nisoldipine + Bergamottin	Grapefruit juice can increase the serum concentrations and oral bioavailability of Nisoldipine due to the inhibitant effect to CYP3A4. Bergamottin is the most abundant of furanocoumarins present in grapefruit juice.	Nisoldipin should not be taken with food containing Bergamottin.	Paine et al. [58], Takanaga et al. [59]
Midazolam + Licofuranocoumarin	Grapefruit juice is contraindicated when taking Midazolam orally since it contains Furanocoumarin compounds that can inhibit CYP3A4. This will increase bioavailability and change the pharmacodynamics of Midazolam, leading to excessive levels of sedation for the patients.	Midazolam should not be taken with food containing Licofuranocoumarin.	Kupferschmidt et al. [60], Goho et al. [61]
Warfarin + Vitamin K1 2,3-epoxide	Vitamin K can make Warfarin less effective, which means that Warfarin could not prevent a dangerous blood clot.	Warfarin should not be taken with food containing Vitamin K1 2,3-epoxide.	Pedersen et al. [13], Johnson et al. [62]
Warfarin + Dimethyl disulfide	Herbs can increase the risk of bleeding if one is taking Warfarin as an anticoagulant. Dimethyl disulfide is one of the components found in herbs.	Warfarin should not be taken with food containing Dimethyl disulfide	Milić et al. [63], Hu et al. [64]

**Table 6 sensors-23-03962-t006:** Comparison of previous methods and ours in terms of advantages, disadvantages, and main results.

Methods-Architectures	Advantages	Disadvantages	Performances
DeepDDI [5] - Deep Neural Network.	Leveraging the structural similarity of food constituents to interacting drugs to predict accurately DFIs.	Predicting DFIs indirectly and may omit some food constituents	Not clearly stated.
FDMine [38] - Graph mining approach.	Harnessing the similarity data from various subnetworks and merging the information on food items and their compound compositions in a homogeneous graph.	Investigating fewer drug compounds. Hard to reproduce.	Highest precision: 0.84
Ours - Simple classification algorithms.	Direct predictions from SMILES descriptions of drugs and food compounds. High reproducibility.	Not built on state-of-the-art architectures.	External test set Precision: 0.9265 to 1.0. Recall: 0.9441 to 1.0. F1-score: 0.9618 to 0.9862.

## Data Availability

Source codes and data are freely available at https://github.com/Henrykaa/drug_food_interactions (accessed on 12 April 2023).

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
