# Peer review of "Development and Validation of an Explainable Machine Learning-Based Prediction Model for Drug–Food Interactions from Chemical Structures"

_sensors, 2023, doi:10.3390/s23083962_

Round 1
Reviewer 1 Report
1. Many spelling and grammar mistakes. The paper should be revised.
2. The contribution of the paper is not clear. Authors should list the contributions in points.
3. A table should be created that summarizes the prior methods that have been presented to study DFI and lists the advantages and disadvantages of each method.
4. In Figure 1, phases 2 and 3 are unclear. Please make the font larger.
5. Is step gathering and preprocessing the same as step extraction and processing? Standardize the expressions if yes.
6. To facilitate understanding, authors should provide descriptions of a sample of the data both before and after processing.
7. Authors are focusing on the classical classifiers in comparison. It is preferable to use DL (encoder-decoder) or more sophisticated ML approaches to evaluate the model.
8. Why do authors utilize VT first, then Pearson correlation, rather than the other way around?
9. Why do authors tune training using randomized search cross-validation?
10.The proposed model should be compared to other model in terms of Precision, Recall and F-measure to study its stability.
11. In addition to Figure 2, authors should include a figure or table summarizing the suggested model results in terms of Precision, Recall, and F-measure.
Author Response
1) Many spelling and grammar mistakes. The paper should be revised.
Answer: We have revised all spelling and grammar mistakes and corrected all where necessary.
2) The contribution of the paper is not clear. Authors should list the contributions in points.
Answer: We have described the contributions in more details, especially in the “Conclusion” section as follows:
“To reduce the number of ADEs due to DFIs and DNIs, we propose a new classification model based on the XGBoost classifier and eighteen optimal features. The ability of our model to predict adverse interactions between drug-food compounds can contribute to drug discoveries and the conformity between prescribed drugs and dietary plans in clinical medicine. From a large number of drug and food constituents (13,580 and 70,477, respectively), we reduced them to only 1,133 drugs and 4,341 food compounds. We also identified the eighteen most important features that yielded the highest predictive performance on DFIs. Each feature represents a chemical reaction between atoms. Thus, our findings could contribute to understanding DFIs at the atomic level along with biological experiments. We believe these findings will benefit the scientific community and patients by decreasing the number of ADEs caused by DFIs and setting a new standard for DFI prediction models.”
3) A table should be created that summarizes the prior methods that have been presented to study DFI and lists the advantages and disadvantages of each method.
Answer: We added Table 6 summarizing the previously proposed methods.
4) In Figure 1, phases 2 and 3 are unclear. Please make the font larger.
Answer: We enlarged the words in stages 2 and 3 as suggested.
5) Is step gathering and preprocessing the same as step extraction and processing? Standardize the expressions if yes.
Answer: No, the two steps were different. First, we downloaded the databases, excluded the samples that could not be used further, and then prepared the training, validation, and testing set. After that, in the “Model building and optimization” stage, we extracted the features and constructed the predictive model. We modified the subsections to make them more understandable.
6) To facilitate understanding, authors should provide descriptions of a sample of the data both before and after processing.
Answer: We have carefully described the changes in the number of drugs, food compounds, and drug-food-compound pairs after pre-processing (please refer to 2.1.2 Data pre-processing, and Results section).
7) Authors are focusing on the classical classifiers in comparison. It is preferable to use DL (encoder-decoder) or more sophisticated ML approaches to evaluate the model.
Answer: We aim to create a simple but robust and reproducible model for everyone to use; therefore, we experimented with straightforward algorithms. We already compared to a DL-based classifier in our experiments, the Multi-layer Perceptron (MLP), and MLP was inferior to the optimal eXtreme Gradient Boosting model.
8) Why do authors utilize VT first, then Pearson correlation, rather than the other way around?
Answer: We tried the methods in different orders. The best performance was when conducting Variance threshold -> Pairwise Pearson’s correlation feature selection -> LASSO -> RFE-CV. Therefore, we proposed our strategy as a new approach to select semantic features in this kind of research.
9) Why do authors tune training using randomized search cross-validation?
Answer: We tried Grid search cross-validation (CV) and Randomized search CV for model finetuning. Still, in the end, we chose the latter because it used much fewer resources (CPU, RAM) in a shorter runtime to generate the results compared to the former.
10) The proposed model should be compared to other model in terms of Precision, Recall and F-measure to study its stability.
Answer: We already compared our model’s performance with other methods (i.e., Ryu et al., 2018; Rahman et al., 2022) in the Results and Discussion section. We added Table 6 as suggested.
11) In addition to Figure 2, authors should include a figure or table summarizing the suggested model results in terms of Precision, Recall, and F-measure.
Answer: We added Table 4 to reveal the model’s performance for each class on four metrics Accuracy, Precision, Recall, and F1-score.
Reviewer 2 Report
Updated and recent references need to be included
Author Response
Many thanks for your constructive comments. In the revised version, we have included more latest references in this field. Please check it again!
Reviewer 3 Report
This paper suggests the use of machine learning to predict drug-food interactions based on their structures from the DrugBank and FooDB. Some comments and suggestions are as follows.
1. The abstract is good and clearly explains what should be presented in the manuscript. The author suggested applying their model to recommend whether a drug should or should not be taken with some foods. However, in the result, there is no section reporting the predicted pairs of drug-food interactions (maybe with some scores) which have not been known before. A list of all predicted drug-food interactions should be provided.
2. In the performance measure part (section 2.7), what do TP, TN, FP, and FN stand for? Please clarify.
3. Since the model was for three class labels, how would the authors measure the performance like the accuracy, recall, precision, and F1-score? Try to clarify and describe them for three-class labels rather than two-class labels.
4. The author explained Precision, Recall, and F1-score in the performance or evaluation section; however, the authors have not used or have not represented them in the result part. The author showed only the accuracy of each ML technique. It would be great if the authors also provide the precision, recall, and F1-score as well.
5. New potential DPIs that were detected by this model should be provided and selected for further explanation to support the possibilities with some knowledge, literature, and theory supported.
Author Response
1. The abstract is good and clearly explains what should be presented in the manuscript. The author suggested applying their model to recommend whether a drug should or should not be taken with some foods. However, in the result, there is no section reporting the predicted pairs of drug-food interactions (maybe with some scores) which have not been known before. A list of all predicted drug-food interactions should be provided.
Answer: Thank you for your question. In Table 4, we only showed recommendations on some commonly used drugs and food compounds reported in the literature. For the interactions of drugs and food compounds that have not been known before, readers can input their names to our web server (mentioned in our future plans in the manuscript) for recommendations. We also provided full datasets in the Github repository.
2. In the performance measure part (section 2.7), what do TP, TN, FP, and FN stand for? Please clarify.
Answer: We modified it as requested.
3. Since the model was for three class labels, how would the authors measure the performance like the accuracy, recall, precision, and F1-score? Try to clarify and describe them for three-class labels rather than two-class labels.
Answer: In the manuscript, we computed the metrics to assess the model’s performance in each class, and they can be used in a three-class task. For example, in Figure 2, the accuracy for class 0 (negative DFIs) equals: [48,315+(92,172+131,359+690+431)]/281,019 = 97.13%. Additionally, please refer to Table 4 (we have just added it as suggested by Reviewer 1) for more details.
4. The author explained Precision, Recall, and F1-score in the performance or evaluation section; however, the authors have not used or have not represented them in the result part. The author showed only the accuracy of each ML technique. It would be great if the authors also provide the precision, recall, and F1-score as well.
Answer: We added Table 4 to reveal the model’s performance for each class on four metrics Accuracy, Precision, Recall, and F1-score.
5. New potential DPIs that were detected by this model should be provided and selected for further explanation to support the possibilities with some knowledge, literature, and theory supported.
Answer: We are not trying to propose any new or potential DFIs in the manuscript. We help patients avoid combining harmful drug-food-constituent pairs, and the ground-truth information to train the model is annotated in DrugBank (from literature evidence).
Round 2
Reviewer 1 Report
the authors addressed my questions.
Reviewer 3 Report
All comments have been addressed.